# Modeling Empathic Similarity in Personal Narratives

**Jocelyn Shen**[1]     **Maarten Sap**[2,3]     **Pedro Colon-Hernandez**[1]
**Hae Won Park**[1]     **Cynthia Breazeal**[1]

[1]Massachusetts Institute of Technology, Cambridge, MA, USA
[2]Carnegie Mellon University, Pittsburgh, PA, USA
[3]Allen Institute for Artificial Intelligence, Seattle, WA, USA
`joceshen@mit.edu, maartensap@cmu.edu, {pe25171, haewon, cynthiab}@mit.edu`

## Abstract

The most meaningful connections between people are often fostered through expression of shared vulnerability and emotional experiences in personal narratives. We introduce a new task of identifying similarity in personal stories based on *empathic resonance*, i.e., the extent to which two people empathize with each others' experiences, as opposed to raw semantic or lexical similarity, as has predominantly been studied in NLP. Using insights from social psychology, we craft a framework that operationalizes empathic similarity in terms of three key features of stories: main events, emotional trajectories, and overall morals or takeaways. We create EMPATHICSTORIES, a dataset of 1,500 personal stories annotated with our empathic similarity features, and 2,000 pairs of stories annotated with empathic similarity scores. Using our dataset, we finetune a model to compute empathic similarity of story pairs, and show that this outperforms semantic similarity models on automated correlation and retrieval metrics. Through a user study with 150 participants, we also assess the effect our model has on retrieving stories that users empathize with, compared to naive semantic similarity-based retrieval, and find that participants empathized significantly more with stories retrieved by our model. Our work has strong implications for the use of empathy-aware models to foster human connection and empathy between people.

## 1 Introduction

Through personal experience sharing, humans are able to feel the sting of another person's pain and the warmth of another person's joy. This process of empathy is foundational in the ability to connect with others, develop emotional resilience, and take prosocial actions in the world (Coke et al., 1978; Morelli et al., 2015; Vinayak and Judge, 2018; Cho and Jeon, 2019). Today, there is more visibility into the lives of others than ever before, yet loneliness

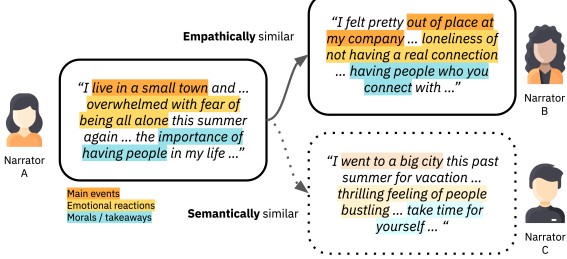

Figure 1: Examples of empathically similar and dissimilar stories. Highlighted are the features of our empathic similarity framework (main event, emotion, and moral/takeaway). Narrator A and B are more likely to empathize with one another over their shared feelings of isolation.

and apathy are widespread (Buecker et al., 2021; Konrath, 2013; Konrath et al., 2011). While these challenges cannot be solved with technology alone, AI systems can be developed to bolster emotional support, empathy, and truly meaningful connections through fostering personal experience sharing (Sagayaraj et al., 2022; Chaturvedi et al., 2023; Berridge et al., 2023). In order to do so, these systems must be able to reason about complex social and emotional phenomena between people.

In this work, we introduce the task of modeling *empathic similarity*, which we define as people's perceived similarity and resonance to others' experiences. For example, in Figure 1, empathic similarity aims to capture that Narrator A, who feels lonely in their small town, is likely to empathize with Narrator B, who is feeling isolated at their new job. Crucially, empathic similarity differs from traditional notions of textual similarity that have been the main focus of NLP work (e.g., semantic similarity; Reimers and Gurevych, 2019); Narrator A will likely not empathize with Narrator C, despite both stories having higher semantic similarity.

We operationalize empathic similarity around alignment in three features of a personal story

(highlighted in Figure 1): its *main event*, its *emotional reaction*, and its overall *moral or story takeaway* (Hodges et al., 2010; Morelli et al., 2017; Krebs, 1976; Wondra and Ellsworth, 2015; Bal and Veltkamp, 2013; Walker and Lombrozo, 2017; Labov and Waletzky, 1997), as motivated by social psychology and narratology literature. From our definition, empathic similarity arises from the interplay of the main events, emotions, and morals in story, where some components or all components must be similar in order for two narrators to resonate with one another. For example, Narrator A and B both experience loneliness, even though their actual situations are different (living in a small town versus working at a company).

To enable machines to model empathic similarity, we introduce EMPATHICSTORIES,[1] a corpus of 1,500 personal stories, with crowdsourced annotations of the free-text summaries of the main event, emotion, and moral of the stories, as well as an empathic similarity score between 2,000 pairs of stories. We find that finetuning on our paired stories dataset to predict empathic similarity improves performance on automatic metrics as compared to off-the-shelf semantic similarity methods.

While automatic evaluation a valuable signal of model quality, it is crucial to showcase the real-world impact of our task on improving empathy towards people's stories. As such, we conducted a full user study with 150 participants who wrote their own personal journal entries and were presented stories retrieved by our model (and by a semantic similarity baseline). Our results show that users empathize significantly more with stories retrieved by our finetuned empathic similarity model compared to those from a semantic similarity baseline (SBERT; Reimers and Gurevych, 2019). Our findings highlight the applicability of our framework, dataset, and model towards fostering meaningful human-human connections by enabling NLP systems to reason about complex interpersonal social-emotional phenomena.

## 2 Related Work

Document similarity is a well-defined task in NLP (Salton et al., 1997; Damashek, 1995; Deerwester et al., 1990; Landauer and Dumais, 1997), but few have applied this work to matching personal

narratives based on shared emotional experiences (Chaturvedi et al., 2018; Lin et al., 2014). One study used Latent Dirichlet Allocation (LDA) to cluster cyberbullying stories and match these stories based on similarity in theme (Dinakar et al., 2012), but discovered that only 58.3% found the matched story to be helpful if provided to the narrator of the original story.

Other work has explored ways to bridge the features of a story and human-perceived similarity of stories (Nguyen et al., 2014). Saldias and Roy (2020) found that people use Labov's action (series of events) and evaluation (narrator's needs and desires) clauses to identify similarity in personal narratives (Labov and Waletzky, 1997). Their findings support our decision to focus on modeling events, emotions, and morals within stories.

Most relevant to our work are recent advances in social and emotional commonsense reasoning using using language models. Specifically, prior methods have used finetuning of language models such as BERT (Devlin et al., 2019; Reimers and Gurevych, 2019) and GPT-2 (Radford et al.) to model events and the emotional reactions caused by everyday events (Rashkin et al., 2019, 2018; Sap et al., 2019b; Bosselut et al., 2019; Wang et al., 2022; West et al., 2022; Mostafazadeh et al., 2020) as well as predicting empathy, condolence, or prosocial outcomes (Lahnala et al., 2022a; Kumano et al.; Boukricha et al., 2013; Zhou and Jurgens, 2020; Bao et al., 2021). Understanding the emotional reactions elicited by events is a challenging task for many NLP systems, as it requires commonsense knowledge and extrapolation of meanings beyond the text alone. Prior works use commonsense knowledge graphs to infer and automatically generate commonsense knowledge of emotional reactions and reasoning about social interactions (Sap et al., 2019c,b; Bosselut et al., 2019; Hwang et al., 2021). However, there are still many under-explored challenges in developing systems that have social intelligence and the ability to infer states between people (Sap et al., 2022).

In contrast to previous works, we present a task for reasoning between pairs of stories, beyond predicting social commonsense features of texts alone. Our work builds on top of prior work by developing a framework around empathic resonance in personal narratives in addition to assessing the human effect of AI-retrieved stories on empathic response beyond automatic metrics. Unlike previous

---

[1]We publicly release our dataset, annotation procedure, model, and user study at `https://github.com/mitmedialab/empathic-stories`

works, our human evaluation is a full user study to see how the model performs given a story that the users told themselves, which is much more aligned with real-world impact.

## 3 Empathic Aspects of Personal Stories

Modeling empathic similarity of stories requires reasoning beyond their simple lexical similarities (see Figure 1). In this section, we briefly discuss how social science scholars have conceptualized empathy (§3.1) and draw on empathy definitions relevant for the NLP domain (Lahnala et al., 2022b). Then, we introduce our framework for modeling *empathic similarity* of stories and its three defining features (§3.2).

### 3.1 Background on Empathy and Stories

Empathy, broadly defined as the ability to feel or understand what a person is feeling, plays a crucial role in human-human connections. Many prior works in social psychology and narrative psychology find that the perceived similarity of a personal experience has effects on empathy (Roshanaei et al., 2019; Hodges et al., 2010; Wright, 2002; Morelli et al., 2017; Krebs, 1976; Wondra and Ellsworth, 2015). For example, Hodges et al. (2010) found that women who shared similar life events to speakers expressed greater empathic concern and reported greater understanding of the speaker.

As with these prior works, our work uses sharing of personal stories as a means to expressing similarity in shared experiences. Personal storytelling as a medium itself has the ability to reduce stress, shift attitudes, elicit empathy, and connect others (Green and Brock, 2000; Andrews et al., 2022; Brockington et al., 2021). In fact, some research has shown that when telling a story to a second listener, speakers and listeners couple their brain activity, indicating the neurological underpinnings of these interpersonal communications (Honey et al., 2012; Vodrahalli et al., 2018).

### 3.2 Empathic Similarity in Personal Stories

We define *empathic similarity* as a measure of how much the narrators of a pair of stories would empathize with one another. While there are many ways to express empathy, we focus specifically on situational empathy, which is empathy that occurs in response to a social context, conveyed through text-based personal narratives (Fabi et al., 2019).

We operationalize an empathic similarity framework grounded in research from social and narrative psychology discussed in §3.1. Our framework differs from prior work (Sharma et al., 2020) in that it is expanded to the relationship between two people's experiences, rather than how empathetically someone responds, and focuses on learning a continuous similarity signal as opposed to detecting the presence of empathy. This distinction is important, as someone may be able to express condolences to a personal experience, but not necessarily relate to the experience itself. The core features of empathic similarity we identify are explained below, and we show how these features contribute to empathic similarity in Appendix A.

**(1) Main event.** Prior work demonstrates that people empathize more with experiences that are similar to their own (Hodges et al., 2010; Morelli et al., 2017; Krebs, 1976). We formalize this as the main event of the story expressed in a short phrase (e.g. "living in a small town").

**(2) Emotional Reaction.** Although two people may relate over an experience, they may differ in how they emotionally respond to the experience (e.g. "overwhelmed with fear of being all alone" vs "loneliness of not having a real connection"). Prior work shows that people have a harder time empathizing with others if they felt that the emotional response to an event was inappropriate (Wondra and Ellsworth, 2015).

**(3) Moral.** Readers are able to abstract a higher-level meaning from the story, often referred to as the moral of the story (Walker and Lombrozo, 2017) (e.g. "the importance of having people around"). In studying fictional narratives, prior work has found that people can empathize with the takeaway of a story, despite its fictional nature (Bal and Veltkamp, 2013).

## 4 EMPATHICSTORIES Dataset

We introduce EMPATHICSTORIES, a corpus of personal stories containing 3,568 total annotations. Specifically, the corpus includes empathic similarity annotations of 2,000 story pairs, and the main events, emotions, morals, and empathy reason annotations for 1,568 individual stories. An overview of our data annotation pipeline is shown in Figure 2 and data preprocessing steps are included in Appendix D. In Appendix H, we show that using LLMs for human annotation is not viable for our task.

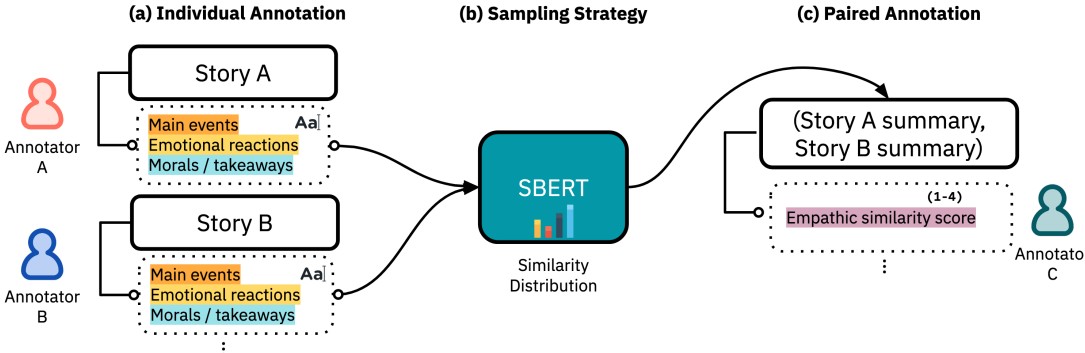

Figure 2: Overview of annotation pipeline starting with (a) individual story event, emotion, and moral to (b) using these annotations to sample balanced story pairs and (c) rating empathic similarity scores

|  | # sents | # words |
|---|---|---|
| **Story** | 13.17 | 235.14 |
| *Main Event* | 1.48 | 32.51 |
| *Emotional Reaction* | 2.39 | 46.08 |
| *Moral* | 1.38 | 31.35 |

Table 1: Story and annotation statistics

| Topic | Keywords | % Stories |
|---|---|---|
| *romantic relationships* | relationships, divorced, passion | 15.63% |
| *positive life events* | opportunities, wedding, cruise | 13.20% |
| *depression* | depression, therapy, psych | 12.95% |
| *family* | families, parents, relatives | 10.33% |
| *substance use* | recovery, drugs, addiction | 9.38% |
| *encouragement* | encouragement, caring, distress | 8.42% |
| *college and school* | students, classes, college | 7.08% |
| *loneliness* | loneliness, relationships, haircut | 5.87% |
| *youth* | teenage, childhood, twenties | 4.97% |
| *life changes* | goodbyes, retired, graduating | 4.40% |
| *work* | mundane, coworkers, volunteering | 4.34% |
| *trauma* | abused, traumas, therapist | 3.44% |

Table 2: Themes across main events of the stories.

## 4.1 Data Sources

We collect a diverse set of stories from sources including social media sites, spoken narratives, and crowdsourced stories. We take approximately 500 stories from each of the following sources (for a full breakdown see Appendix F). These sources contain English-written stories revolving around deep emotional experiences and open-ended conversation starters.

**(1) Online Personal Stories.** We scrape stories from subreddits[2] about personal experiences (*r/offmychest*, *r/todayiamhappy*, and *r/casualconversation*). We also include a small set of stories from a public college confessions forum.

**(2) Crowdsourced Personal Stories.** We use a subset of autobiographical stories from the existing Hippocorpus dataset (Sap et al., 2020), which contains recalled and imagined diary-like personal stories obtained from crowdworkers.

**(3) Spoken Personal Narratives.** We use stories from the Roadtrip Nation corpus (Saldias and Roy, 2020), which contains transcribed personal stories about people's career trajectories and life stories.

## 4.2 Individual Story Annotation

Using these stories, we designed an annotation framework on Amazon Mechanical Turk (MTurk) that asks workers to label individual story features. Then, we asked for short free responses on (1) the

---

[2] https://api.pushshift.io/

main event of the story, (2) the main emotional state induced by the main event, and (3) moral(s) of the story. The story and annotated summary statistics are shown in Table 1. The themes from stories are shown in Table 2, and themes for annotated summaries as well as our topic modeling approach are presented in Appendix E.

## 4.3 Paired Story Annotation

**Sampling Empathic Story Pairs.** We devise a sampling method to create a sample of balanced empathically similar and dissimilar story pairs, since random sampling across all possible pairs would likely result in an unbalanced dataset with more dissimilar stories than similar stories. First, we split the 1,568 stories into a train, dev, and test set using a 75/5/20 split. Using SBERT (Reimers and Gurevych, 2019), we compute a composite similarity score using average cosine similarity of the embeddings for the story and our 3 empathy features for every possible story pair within the dataset. We randomly sample stories from each bin such that bins with higher composite similarity scores are more likely to be chosen.

**Annotation Procedure** With the sampled story pairs, we released an annotation task on Amazon

| Annotation | | PPA | KA |
|---|---|---|---|
| Empathic similarity | **Overall** | .80 | .14 |
| | Train | .79 | .14 |
| | Dev | .81 | .11 |
| | Test | .83 | .17 |
| Event similarity | **Overall** | .86 | .27 |
| | Train | .86 | .26 |
| | Dev | .84 | .25 |
| | Test | .87 | .30 |
| Emotion similarity | **Overall** | .83 | .23 |
| | Train | .83 | .23 |
| | Dev | .79 | .15 |
| | Test | .84 | .25 |
| Moral similarity | **Overall** | .80 | .19 |
| | Train | .80 | .18 |
| | Dev | .80 | .14 |
| | Test | .82 | .20 |

Table 3: Similarity agreement scores (PPA = pairwise percent agreement, KA = Krippendorff's Alpha)

MTurk, asking workers to read pairs of stories and rate various aspects of empathic similarity between the stories. Two annotators rated each story pair. From early testing, we found that the task was difficult because of the large amount of text in the stories and the cognitive load of projecting into two narrator's mental states. To simplify the task, we used ChatGPT (`gpt-3.5-turbo`) to summarize all the stories before presenting the pairs to annotators. While summarization may remove specific details of the stories, we find that the main event, emotion, and moral takeaway are still present.[3]

At the beginning of the task, we first provide the annotator with 6 examples of empathically similar stories: one positive and one negative example for stories that are empathically similar/dissimilar based on each feature: main event, emotion, and moral of the story. After reading the two stories, we ask workers to provide explanations of whether and why the narrators would empathize with one another, to prime annotators to think about the empathic relationship between the stories. We then ask workers to provide four similarity ratings on a 4-point Likert scale (1 = strongly disagree, 4 = strongly agree): (1) overall empathic similarity (how likely the two narrators would empathize with each other), (2) similarity in the main events, (3) emotions, and (4) morals of the stories.

**Agreement** We aggregate annotations by averaging between the 2 raters. Agreement scores for em-

---

[3]By comparing the cosine similarity of human annotated event, emotion, and moral to the ChatGPT summarized stories, we find that there is high semantic overlap of the human ground-truths to the automatically generated summaries (0.66 for event, 0.64 for emotion, and 0.49 for moral).

pathy, event, emotion, and moral similarity across the entire dataset are shown in Table 3. While these agreement scores are seemingly on the lower side, using a softer constraint, we see that most common disagreements are at most 1 likert point away (73% of points are at most 1 distance away). We are aiming for a more descriptive annotation paradigm and thus do not expect annotators to perfectly agree (Rottger et al., 2022). Furthermore, our agreement rates are in line with other inherently personal and affect-driven annotation tasks (Sap et al., 2017; Rashkin et al., 2018). Given the difficulty of our task (reading longer stories and projecting the mental state of 2 characters), our agreement is in line with prior work, which achieve around 0.51 - 0.91 PPA and 0.29 - 0.34 KA.

## 5 Modeling Empathic Similarity

To enable the retrieval and analysis of empathically similar stories, we design a task detailed below. In Appendix B, we also propose an auxiliary reasoning task to automatically extract event, emotion, and moral features from stories, which could be used in future work to quickly generate story annotations.

### 5.1 Task Formulation

Our ultimate retrieval task is given a query story $Q$ and selects a story $S_i$ from a set of $N$ stories $\{S_1, S_2, ..., S_N\}$ such that $i = argmax_i \ sim(f_\theta(S_i), f_\theta(Q))$. Here, $sim(\cdot, \cdot)$ is a similarity metric (e.g. cosine similarity) between two story representations $f_\theta(S_i)$ and $f_\theta(Q)$ that are learned from human ratings of empathic similarity.

**Empathic Similarity Prediction.** The overall task is, given a story pair $(S_1, S_2)$, return a similarity score $sim(f_\theta(S_i), f_\theta(Q))$ such that $sim(\cdot, \cdot)$ is large for empathically similar stories and small for empathically dissimilar stories.

### 5.2 Models

We propose finetuning LLMs to learn embeddings that capture empathic similarity using cosine distance, for efficient retrieval at test time. In contrast, a popular approach is to use few-shot prompting of very large language models (e.g., GPT-3 and ChatGPT), which have shown impressive performance across a variety of tasks (Brown et al., 2020). However, in a real deployment setting, retrieval through prompting every possible pair of stories is expensive and inefficient.

| Model | $r$ | $\rho$ | Acc | $P$ | $R$ | $F1$ | $P_{k=1}$ | $\tau_{rank}$ | $\rho_{rank}$ |
|---|---|---|---|---|---|---|---|---|---|
| SBERT | 30.93 | 29.86 | 62.75 | 57.81 | 90.24 | 70.48 | 57.92 | 17.46 | 18.74 |
| + finetuning | **35.93** | **35.21** | **64.75** | 58.68 | 90.73 | **71.26** | 57.43 | 17.59 | 18.98 |
| BART | 10.24 | 11.54 | 57.00 | 52.19 | **99.02** | 68.35 | 49.51 | 7.56 | 9.28 |
| + finetuning | 34.20 | 34.43 | **64.75** | 58.2 | 88.29 | 70.16 | 65.84 | **24.68** | **26.55** |
| GPT-3 | 3.24 | 2.79 | 51.25 | 51.25 | 100 | 67.77 | **90.59** | 0.33 | 0.79 |
| + 5 examples | 4.94 | 6.71 | 51.25 | 51.27 | 98.54 | 67.45 | 72.77 | -4.8 | -5.33 |
| ChatGPT | 19.56 | 20.16 | 56.25 | 55.24 | 77.07 | 64.36 | 80.69 | 13.48 | 14.10 |
| + 5 examples | 27.75 | 28.07 | 63.25 | **60.43** | 81.95 | 69.57 | 85.15 | 21.27 | 22.10 |

Table 4: Model performance for empathic similarity prediction task across correlation, accuracy, and retrieval metrics. $r$ = Pearson's correlation, $\rho$ = Spearman's correlation, Acc = accuracy, $P$ = precision, $R$ = recall, $P_{k=1}$ = precision at $k$ where $k$ is 1, $\tau_{rank}$ = Kendall Tau of ranking and $\rho_{rank}$ = Spearman of ranking. Note that all scores are multiplied by 100 for easier comparison, and the maximum for each metric is 100. In **bold** is the best performing and underlined is the second-best performing condition for the metric.

**Baseline Models.** We compare performance to finetuning with SBERT (multi-qa-mpnet-base-dot-v1) (Reimers and Gurevych, 2019; Brown et al., 2020) and BART model (bart-base) (Lewis et al., 2019). As a few-shot baseline, we evaluate GPT-3 (text-davinci-003) and ChatGPT's (gpt-3.5-turbo) ability to distinguish empathically similar stories by using a $k$-shot prompting setup as done in Sap et al. (2022); Brown et al. (2020). For the query story pair, we ask for an empathic similarity score from 1-4. We compare across $k = 0$ examples and $k = 5$ examples from the training set. We also evaluate these models' ability to generate human-like main event, emotion description, and moral summaries for each story. Again, we use a $k$-shot prompting setup, comparing across $k = 0$ and $k = 10$ examples. See Appendix G and Appendix C for prompts used and finetuning details.

**Empathy Similarity Prediction.** We propose a bi-encoder architecture finetuned with mean-squared error (MSE) loss of the cosine-similarity between story pairs, as compared to the empathic similarity gold labels. For each of the encoders, we use a shared pretrained transformer-based model and further finetune on the 1,500 annotated story pairs in our training set. We obtain the final embedding using mean pooling of the encoder last hidden state.

## 6 Automatic Evaluation

To evaluate the quality of empathic similarity predictions, we first compare the Spearman's and Pearson's correlations between the cosine similarity of the sentence embeddings and the gold empathic similarity labels. Next, we bin scores into binary similar/dissimilar categories ($> 2.5$ and $\leq 2.5$ respectively) compute the accuracy, precision, recall, and F1 scores. Finally, we compute a series of retrieval-based metrics including precision at

$k = 1$ (what proportion of the top-ranked stories by our model are the top-ranked story as rated by human annotators), Kendall's Tau (Abdi, 2007), and Spearman's correlation (Schober et al., 2018) for the ranking of the stories (how close the overall rankings are).

Shown in Table 4, our results indicate that finetuning SBERT and BART with EMPATHICSTORIES results in performance gains across all metrics. SBERT has relatively high off-the-shelf performance, as it is trained with 215M examples specifically for semantic similarity tasks. However, we see that finetuning with our dataset, which contains far fewer training examples relative to SBERT's pretraining corpus, improves performance. (+ 5.35 $\rho$, +2 accuracy). BART, which is not specifically pre-trained for semantic similarity tasks, shows even greater gains across retrieval metrics when finetuned on our dataset. (22.89 $\rho$, +7.75 accuracy). We find that for BART models, fine tuning improvements ($p = 0.02$, $p = 0.0006$ respectively), as measured with McNemar's test on the accuracy scores and Fisher's transformation on correlations, are significantly higher than baselines.

While GPT-3 and ChatGPT have high performance on the precision at $k$ retrieval metric, in practice, it is not feasible to prompt the models with every pair of stories in the retrieval corpus.

## 7 User Study

Prior work's versions of human evaluations (Zhou and Jurgens, 2020; Bao et al., 2021; Sharma et al., 2020) are humans verifying or ranking model outputs based on inputs from test data. This provides a valuable signal of model quality, but isn't representative of how a model could be used in real-world applications due to input distribution mismatch and lack of personal investment in the task. Our hu-

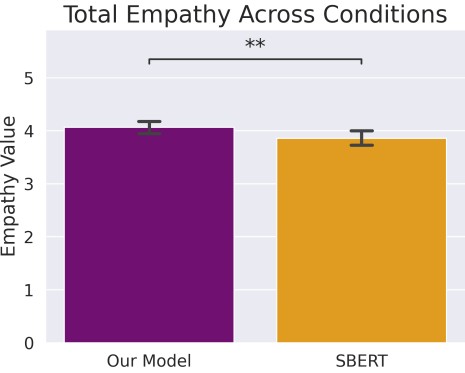

Figure 3: Total empathy for the story retrieved by our model vs. SBERT. Error bars show standard error.

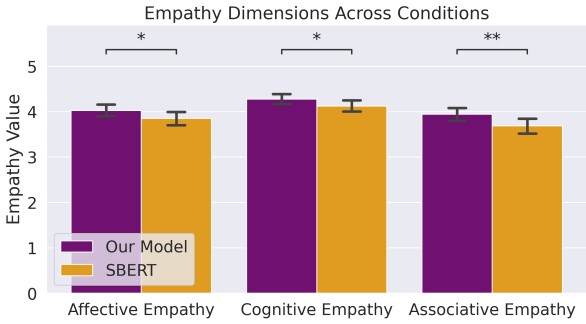

Figure 4: Breakdown of empathy dimensions for the story retrieved by our model vs. SBERT

man evaluation is a full user study to see how the model performs in retrieving a story that is empathically similar to a story that the users told themselves. Through our user study, we demonstrate the applicability of the task to improve empathy towards retrieval of human stories, as well as how our dataset was used to develop the empathic similarity retrieval task and why the task matters in the real-world. Our hypothesis is: *Users will empathize more with stories retrieved by our model (BART finetuned on* EMPATHICSTORIES*) than stories retrieved by SBERT.*

### 7.1 Participants and Recruitment

We recruited a pool of 150 participants from Prolific. Participants were primarily women (58%, 38% men, 3% non-binary, 1% undisclosed) and white (73%, 8% Black, 9% other or undisclosed, 4% Indian, 3% Asian, 2 % Hispanic, 1% Native American). The mean age for participants was 37 (s.d. 11.6), and participants on average said they would consider themselves empathetic people (mean 4.3, s.d. 0.81 for Likert scale from 1-5).

### 7.2 Study Protocol

Participants rated their mood, wrote a personal story, then rated their empathy towards the stories retrieved by the baseline and proposed models. They additionally answered questions about the story they wrote (main event, emotion, and moral of the story) and their demographic information (age, ethnicity, and gender).

**User Interface.** We designed a web interface similar to a guided journaling app and distributed the link to the interface during the study. The interface connects to a server run on a GPU machine

(4x Nvidia A40s, 256GB of RAM, and 64 cores), which retrieves story responses in real time.

**Writing Prompts and Stories Retrieved.** We carefully designed writing prompts to present to the participants to elicit highly personal stories, inspired by questions from the Life Story Interview (McAdams, 2007), an approach from social science to gather key moments from a person's life.

**Conditions.** We used a within-subject study design, where each participant was exposed to 2 conditions presented in random order. In Condition 1, participants read a story retrieved by our best performing model on the empathic similarity task (BART + finetuning). In Condition 2, participants read a story retrieved by SBERT. For both models, we select the best response that minimizes cosine distance.

**Measures.** To measure empathy towards each story, we used a shortened version of the State Empathy Survey (Shen, 2010), which contains 7 questions covering affective (sharing of others' feelings), cognitive (adopting another's point of view), and associative (identification with others) aspects of situational empathy. We also ask users to provide a free-text explanation of whether and why they found the retrieved story empathically resonant, to gain qualitative insights into their experience.

### 7.3 Effects on Empathy

With our results shown in Figure 3, we found through a paired t-test ($N = 150$) that **users significantly empathized more with stories retrieved by our model finetuned on** EMPATHICSTORIES **than off-the-shelf SBERT** ($t(149) = 2.43$, $p < 0.01$, Cohen's $d = 0.26$), validating our hypothesis. In addition, this effect was present across all three dimensions of empathy: affective ($t(149) = 1.87$, $p = 0.03$, Cohen's $d = 0.21$), cognitive ($t(149) =$

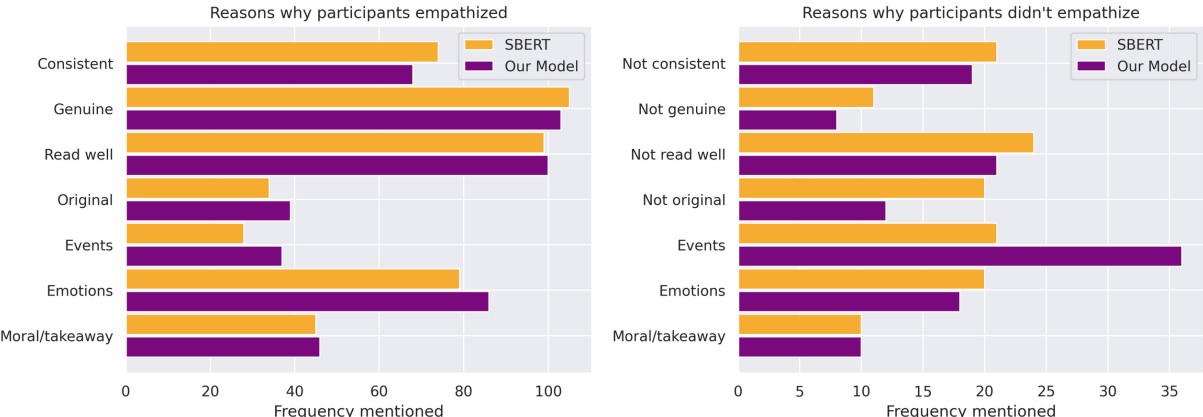

Figure 5: Reasons why participants did or did not empathize with the retrieved story.

2.05, $p = 0.02$, Cohen's $d = 0.21$), and associative empathy ($t(149) = 2.61$, $p = 0.005$, Cohen's $d = 0.27$), as shown in Figure 4 (empathy values are the summed scores from the empathy survey). Interestingly, the difference in empathic response across conditions is strongest for associative empathy, which measures how much the user can identify with the narrator of the story.

We examine reasons why users empathized with retrieved stories across conditions (Figure 5). Across both conditions, empathy towards a story was often related to how well-read, genuine, and consistent the story was, and if the user could empathize with the narrator's emotional reactions. When participants did not empathize with a retrieved story, this was more often than not due to stark differences in the main events of their own story and the model's selected story. This effect was strongest for our finetuned model, as it was trained on data with a more open definition of empathy than just sharing the same situation. In certain cases, this could result in the events being too different for the user to empathize with.

Interestingly, we see that our model chose stories that aligned better on events and emotions with respect to the story they wrote, and participants thought the stories were more original compared to SBERT-retrieved stories. In cases where the participant did not empathize with the retrieved story, SBERT-retrieved stories were considered less consistent, less genuine, less, original, did not read as well, and did not match on emotions as well compared to our model.

From qualitative responses, we see that our model retrieved stories that user empathized with based on the situation described, the emotions the narrator felt, and the takeaway of the story. For example, one participant shared that *"I found no moment where I didn't fully understand the author, and I share a very similar story about my father...its absolutely amazing...I enjoyed this study very much."* Other participants wrote, *"I empathize heavily with this story because it has many similarities to my own. Kind of a 'started from the bottom, now we're here' vibe, which I love to see"* and *"I can relate to the feelings of abandonment and regret expressed."*

## 8 Future Directions for Empathic Similarity

In summary, few prior works on text-based empathy have looked at modeling empathy in two-way interpersonal settings for human-to-human connection, as most focus on detecting empathy or generating empathetic utterances, and even fewer of these works have shown tangible outcomes in human studies. With increasing polarization, loneliness, and apathy (Buecker et al., 2021), personal experiences are a fundamental way people connect, yet existing social recommendation is not targeted for human-human connectivity and empathy. Empathically encoded story embeddings could be useful for a variety of NLP tasks, including retrieval, text generation, dialogue, and translation, for example in the following settings:

- Using empathic reasoning to incorporate story retrieval in dialogue generation.

- Generating stories that users resonate with more in conversational AI

- Extending this work to multilingual settings and better understand translating experiences

in ways that preserve empathic meaning

- Better understand cognitive insights, such as linguistic patterns of emotion-driven communication

- Applications and building interactions that foster story sharing across geographic, ethnic, and cultural bridges, such as developing better social media recommendation or personalization.

We encourage future works to explore these directions in developing more human-centered approaches for interactions with NLP systems.

## 9  Conclusion

This work explores how we can model empathic resonance between people's personal experiences. We focused specifically on unpacking empathy in text-based narratives through our framework of the events, emotions, and moral takeaways from personal narratives. We collected EMPATHICSTORIES, a diverse dataset of high-quality personal narratives with rich annotations on individual story features and empathic resonance between pairs of stories. We presented a novel task for retrieval of empathically similar stories and showed that large-language models finetuned on our dataset can achieve considerable performance gains in our task. Finally, we validated the real-world efficacy of our BART-finetuned retrieval model in a user study, demonstrating significant improvements in feelings of empathy towards stories retrieved by our model compared to off-the-shelf semantic similarity retrieval.

Empathy is a complex and multi-dimensional phenomenon, intertwined with affective and cognitive states, and it is foundational in our ability to form social relationships and develop meaningful connections. In a world where loneliness and apathy are increasingly present despite the numerous ways we are now able to interact with technology-based media, understanding empathy, developing empathic reasoning in AI agents, and building new interactions to foster empathy are imperative challenges. Our work lays the groundwork towards this broader vision and demonstrates that AI systems that can reason about complex interpersonal dynamics have the potential to improve empathy and connection between people in the real-world.

## Limitations

With regards to our data collection and annotation framework, our annotations for empathic similarity are not first-person, which are sub-optimal given that it may be difficult for annotator's to project the emotional states of two narrators. In addition, because of the complexity of our annotation task, we opted to use ChatGPT summaries of the stories during our paired story annotation, which could introduce biases depending on the quality of the generated summaries. However, given the inherent difficulty of the task, we found this reduction necessary to achieve agreement and reduce noise in our dataset, and we found that important features will still present in the summaries. Future work could use our human experimental setup to collect first person labels over the entire stories, rather than the automatic summaries.

Another limitation of our modeling approach is that our finetuned model takes in data that captures empathic relations across our framework of events, emotions, and morals. However, the learned story representations are general purpose and are not personalized to a user's empathic preferences. Personalization could improve model performance across automatic and human evaluation metrics, as there may exist finer-grained user preferences in how users empathize with certain stories, and what aspects users focus on. Furthermore, future work could explore training using a contrastive setup to learn more contextualized story embeddings.

Lastly, future work should explore longitudinal effects of recieving stories retrieved by our system. Our survey measures (State Empathy Scale) are used for short, quick assessments of immediate empathy rather than "fixed" or "trait" empathy. While our model might perform well in this one-shot interaction settings, it is also important to study the last empathic effects of reading stories retrieved by the model and measure changes in a user's longer term empathy, mood, and feelings of connection.

## Ethics Statement

While such a system might foster empathy and connectedness, it is important to consider the potential harms brought about by this work. As with many recommenders, our model is susceptible to algorithmic biases in the types of stories it retrieves, as well as creating an echo chamber for homogeneous perspectives (Kirk et al., 2023). Embedding diversity in the recommended stories is important in both

broadening the perspective of users and preventing biases.

Many social platforms struggle with the issue of content moderation and content safety. In its proposed state, our model does not do anything to guarantee the safety of content that is shared with users. Hateful speech and triggering experiences should not be propagated by our model regardless of the extent to which users relate to these stories (Goel et al., 2023; Lima et al., 2018).

Finally, the goal of our work is to connect people to other human experiences. Story generation and NLG that aims to mimic or appropriate human experiences is not something we endorse, and we encourage the use of machine-text detectors in systems that retrieve empathic stories. In line with Oren Etzioni (2018)'s three rules of AI, we also discourage presenting humans with machine-generated stories without disclosing that the story is written by an AI author.

## Acknowledgements

We would like to thank all of our participants, annotators, and teammates for their invaluable contributions to this project. Special thanks to Sharifa Algohwinem and Wonjune Kang for their technical feedback throughout the project and thanks to Ji Min Mun, Akhila Yerukola, and Ishaan Grover for paper feedback. This work was supported by an NSF GRFP under Grant No. 2141064 and the IITP grant funded by the Korean Ministry of Science and ICT No.2020-0-00842.

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

# A  Understanding Aspects of Empathic Similarity

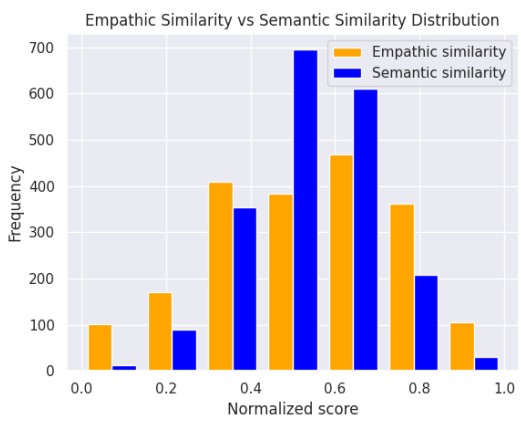

Figure 6: Comparing the empathic similarity and semantic similarity core distributions

Before training any models to learn empathic similarity ratings, it is important to understand the mechanisms behind empathic similarity in text-based personal narratives. In particular, we are interested in how structural elements of stories (events, emotional trajectories, and morals) relate to empathy. The question we aim to answer through our analysis of the text is what qualities of personal experiences people resonate with most and how does this relate to the personal experience they self disclose.

First, we look at the correlation between human-rated similarity in event, emotion, and moral of the stories to the empathic similarity rating. We show in Table 5 that the correlation of the similarity between events, emotions, and morals to the empathic similarity rating is high for all three features. This indicates that similarity in these components is related to similarity in empathic resonance between stories. Using a paired t-test between high and low empathically similar story pairs, we find that empathically similar story pairs have statistically significantly higher similarities in events, emotions, and morals, with the largest increase in moral similarity and roughly equivalent increases in event and emotion similarities.

Next, we look at the differences between semantic similarity and human-rated empathic similarity. As shown in Figure 6, we can see that the distributions of similarity scores are different for human-rated empathic similarity scores as compared to semantic similarity scores obtained from SBERT. Semantic similarity of stories is

| Feature | $r$ | $\rho$ |
|---|---|---|
| Similarity in Main Event | 0.69 | 0.69 |
| Similarity in Emotion Description | 0.65 | 0.65 |
| Similarity in Moral | 0.76 | 0.76 |

Table 5: Correlation between similarity scores for individual features compared to overall empathic similarity score. $r$ = Pearson's correlation coefficient. $\rho$ = Spearman's correlation coefficient.

| Feature | Model | BLEU | ROUGE | METEOR | BertScore |
|---|---|---|---|---|---|
| Event | BART | 1.16 | 16.87 | 21.26 | 13.30 |
| | + finetuning | **9.56** | **32.72** | 29.14 | **39.79** |
| | GPT-3 | 1.40 | 24.77 | 26.31 | 33.39 |
| | + 10 examples | 7.72 | 32.22 | 23.60 | 36.84 |
| | ChatGPT | 1.85 | 25.35 | 25.36 | 34.93 |
| | + 10 examples | 7.23 | 30.02 | **32.81** | 37.59 |
| Emotion | BART | 0.40 | 15.73 | 16.95 | 6.53 |
| | + finetuning | **2.08** | **26.61** | 23.54 | 26.24 |
| | GPT-3 | 1.56 | 22.37 | 27.90 | 21.28 |
| | + 10 examples | 0.08 | 21.09 | 12.08 | 19.97 |
| | ChatGPT | 1.66 | 23.21 | **29.62** | 22.19 |
| | + 10 examples | 1.09 | 25.43 | 27.67 | **26.46** |
| Moral | BART | 0.02 | 11.78 | 15.52 | 0.40 |
| | + finetuning | **13.77** | **33.52** | **29.66** | 32.26 |
| | GPT-3 | 5.86 | 28.10 | 27.87 | 31.64 |
| | + 10 examples | 4.38 | 28.63 | 18.97 | 28.15 |
| | ChatGPT | 4.45 | 25.03 | 26.16 | 30.99 |
| | + 10 examples | 6.63 | 27.91 | 27.51 | **33.97** |

Table 6: Quality of event, emotion, and moral summaries across models. Scores are multiplied by 100 for readability, and the max. for each metric is 100.

weakly positively correlated with empathic similarity ($\rho = 0.17$), with event-based features correlating the most ($\rho = 0.067$), followed by emotion-based features ($\rho = 0.0069$) and lastly moral features ($\rho = -0.048$). These results indicate that semantic similarity is naturally related to empathic similarity, but might not capture relationships between emotions and takeaways in pairs of stories.

# B  Empathy Reasoning Task

**Empathy Reasoning Task Definition.** Given a story context $c$, we finetune a sequence-to-sequence (seq2seq) model to generate an event ($v$), emotion ($e$), and moral ($m$), concatenating annotated summaries to construct the gold label and modeling $p(v, e, m|c)$ (Kim et al., 2022). The model is trained to minimize negative log likelihood of predicting each word in the constructed gold label.

**Empathy Reasoning Results.** We evaluate empathy reasoning performance using BLEU (Papineni et al., 2002), ROUGE (Lin, 2004), METEOR (Banerjee and Lavie, 2005), and BertScore (Zhang et al., 2020), taking the human-written free-text annotations as gold references. From Ta-

ble 6, we see that finetuning BART with human-written story summaries improves performance across all metrics. The BART model finetuned on EMPATHICSTORIES demonstrates improved performance across 3/4 metrics in event and moral reasons. For emotion reasons, ChatGPT demonstrates better performance in 2/4 metrics, with the finetuned BART model close behind. We note that the BART-base model has 140M parameters, whereas ChatGPT has upwards of 175B parameters.

## C Finetuned Model Training Details

We use a 75:5:20 train:dev.:test split on both individual stories and pairs of stories. For the empathic similarity prediction task, we use learning rates of 1e-6 and 5e-6 for SBERT and BART respectively, and a linear scheduler with warmup. For the empathic reasoning task, we use a learning rate of 1e-5. For both tasks, we use a batch size of 8 and finetune for 30-50 epochs, monitoring correlation and validation loss to select the best-performing models. We trained all models on 4x Nvidia A40s with 256GB of RAM and 64 cores, and all model training times were under 12 hours.

## D Data Pre-Processing

For all of the data sources, we remove stories that are shorter than 5 sentences long, longer than 500 words, and which have a severe toxicity score of less than 0.005 using Detoxify (Hanu and Unitary team, 2020). While the latter step may filter out meaningful stories and introduce bias in the story selections (Sap et al., 2019a), we err on the side of removing any stories that could be potentially harmful, even if not severely so.

Our research team then selected stories that were appropriate to share (did not contain excessive profanity or explicit sexual content), and which had a first-person narrator and concrete resolution to the story. We chose stories with a concrete resolution in order to avoid rant posts, which were common on social media pages. In addition, we manually corrected overt grammatical errors as well as references to the platform the story was shared on (e.g. addressing Redditors). Our final set of stories contains 1,568 curated, high-quality personal narratives.

## E Story and Annotation Themes

Below we show the top themes across each story's emotion (Table 7) and moral (Table 8) annotations.

Note that we did not include topics for the events since these were similar to Table 2. To identify these topics, we use Latent Dirichlet Allocation (LDA) and KeyBERT on the clusters (Grootendorst, 2020).

| Topic | Keywords | % Stories |
|---|---|---|
| *depression* | melancholy, depression, unhappy | 28.95% |
| *happiness and satisfaction* | happiness, satisfaction, overwhelmed | 20.92% |
| *anxiety* | anxiety, frustrated, upset | 11.03% |
| *motivation* | motivated, success, achieving | 10.40% |
| *compassion* | compassionate, happiness, gradchildren | 9.38% |
| *gratitude* | gratitude, generosity, happiness | 9.31% |
| *desire* | desire, passion, youth | 6.12% |
| *grief* | grief, sober, lifestyle | 3.89% |

Table 7: Themes across emotion descriptions of the stories.

| Topic | Keywords | % Stories |
|---|---|---|
| *motivation and encouragement* | motivation, success, achieving | 40.31% |
| *overcoming and resilience* | overcome, resilient, rehab | 25.57% |
| *happiness and fulfilment* | opportunities, happiness, meaningful | 17.60% |
| *social support and gratitude* | companionship, gratitude, stress | 16.52% |

Table 8: Themes across morals of the stories.

## F Collected Stories Breakdown

A breakdown of the amount of stories per source can be found in Table 9.

| Data Source | Number of Stories |
|---|---|
| *Hippocorpus* | 483 |
| *Road Trip Narratives* | 476 |
| *Reddit - Today I Am Happy* | 198 |
| *Reddit-Casual Conversations* | 195 |
| *Reddit-Off My Chest* | 162 |
| *Facebook - [Redacted] Confessions* | 54 |

Table 9: Breakdown of retrieved stories per data source.

## G GPT-3 and ChatGPT Prompts

Below are prompts we fed to GPT-3 and ChatGPT for our few-shot baselines. Note that in addition to the prompts, we provided sampled examples from our training corpus.

- **Event summary:** *What is the main event being described in the story? Response must be at least 1 sentence and 50-1000 characters including spaces.*

- **Emotion summary:** *Describe the emotions the narrator feels before and after the main event and why they feel this way. Answer as though you were explaining how the narrator felt to someone who knew nothing about the situation. Response must be at least 2*

*sentences and 150-1000 characters including spaces.*

- **Moral summary:** *What is the high-level lesson or takeaway (ie. moral) of the story? Response must be at least 1 sentence and 100-1000 characters including spaces.*

- **Empathic similarity:** *Rate the extent to which you agree with the statement "the narrators of the two stories would empathize with each other." We define empathy as feeling, understanding, and relating to what another person is experiencing. Note that it is possible to have empathy even without sharing the exact same experience or circumstance. Importantly, for two stories to be empathetically similar, both narrators should be able to empathize with each other (if narrator A's story was shared in response to narrator B's story, narrator B would empathize with narrator A and vice versa). Give your answer on a scale from 1-4 (1 - not at all, 2 - not so much, 3 - very much, 4 - extremely)*

## H  Using LLMs as a Proxy for Human Annotations

Recent works raise the question of whether LLMs can be used to proxy human annotations (Gilardi et al., 2023). The motivation behind this method is that obtaining human labels across many pairs of stories is costly, and this cost only compounds as the number of stories in the corpus increases. As such, we provide additional analyses as to whether or not these models can truly perform at the same level as human annotators for our task, which involves heavy empathy and emotion reasoning.

### H.1  Individual Story Annotation

We prompt ChatGPT (gpt-3.5-turbo) to generate summaries of each story's main event, emotion, and moral, in addition to a list of reasons why a narrator might empathize with the story. We compare these summaries against human-written summaries using BLEU, ROUGE, METEOR, and BertScore (Table 10), showing that ChatGPT has relatively low performance across all four metrics.

### H.2  Paired Story Annotation

We feed the same prompt given to human annotators into ChatGPT, asking for a Likert score from

| Summary | BLEU | ROUGE | METEOR | BertScore |
|---|---|---|---|---|
| Main Event | 2.86 | 26.37 | 28.20 | 36.53 |
| Emotion Description | 1.43 | 23.01 | 28.87 | 23.36 |
| Moral | 7.67 | 27.64 | 27.33 | 33.24 |

Table 10: Quality of ChatGPT story empathy reasoning annotations (scores are multiplied by 100 for readability, and the maximum for each metric is 100)

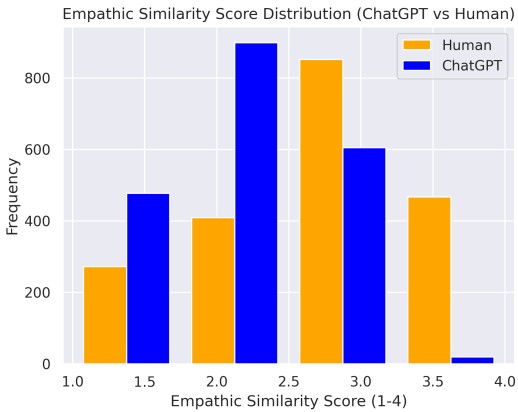

Figure 7: Comparing the empathic similarity score distributions between ChatGPT and human labels

1-4 for the empathic similarity between two stories. The Spearman's correlation between human and ChatGPT generated labels is $0.22$ ($p < 0.001$), indicating weakly positive correlation between human annotations and ChatGPT annotations. In addition, we perform a one-sample t-test on the mean-squared error between automatically generated labels and human annotations across all story pairs in the training data, obtaining a p-value $< 0.001$, indicating that the mean of all the errors is nonzero with statistical significance.

Finally, we bin the ChatGPT annotations into agree/disagree categories, and compute the classification precision (0.59), recall (0.40), F1 score (0.48), and accuracy (0.59) as compared to human gold labels. These scores offer insight as to how well ChatGPT predicts the direction of the empathic similarity annotation, but we see that accuracy is low when comparing to human labels. In Figure 7, we see that ChatGPT similarity scores are skewed to the left, indicating that humans are more likely to find empathic similarities between experiences. These results are also supported by the higher number of false negatives when comparing ChatGPT classification to human gold labels.