# OpenReview forum: "Modeling Empathic Similarity in Personal Narratives"
_EMNLP/2023/Conference — EMNLP 2023 Main_

### Official Review · Reviewer_Pwhy · 2023-08-05

**Soundness:** 4

**Excitement:**

4: Strong: This paper deepens the understanding of some phenomenon or lowers the barriers to an existing research direction.

**Paper Topic And Main Contributions:**

This paper addresses empathic resonance between people, by utilizing their personalized narratives.  They compare empathy similarity through people's stories by using three components of the stories: main event, emotional reaction, and moral. The authors introduce a new dataset, which contains empathic similarity annotations between pairs of stories. Moreover, they introduce two tasks: empathic similarity prediction and empathy reasoning. The main models that they utilize are SBERT, BART (with fine-tuning), and GPT3, ChatGPT (with few-shot learning). In addition, to the automatic evaluation, the authors also conduct a user study where the main goal is to evaluate the models for story retrieval.

The main contributions of the paper are as follows:
- They introduce a framework to model empathic similarity, which focuses on the relationship between the narratives of a pair of people.
- They release “EmpathaticStories”  a corpus of 1500 stories with data from three different sources.
- A user study that involves personalized human evaluation of the models predictions

**Reasons To Accept:**

1. The collected dataset, is a resource that might prove valuable for future work on empathic analysis by utilizing personalized narratives of the users
2. A user study that conducts a personalized evaluation of the model's ability to retrieve similar stories that the user can empathize. I found this user study particularly interesting.

**Reasons To Reject:**

1. The paper mentions that it summarizes stories before presenting the pairs to the annotators. However, valuable information might be lost in the process. For example, the individual annotations of the first part of the annotation, like main events, emotional reaction, or moral, which are an important part of the framework defined for empathy similarity
2. In addition, there is a very low agreement between the annotators, and it is not stated how the annotations are aggregated, or how the disagreements are solved
3. Analysis for the empathy reasoning task is vague. It does not include any human evaluation and no further analysis except reporting automated metrics.

**Reproducibility:**

5: Could easily reproduce the results.

**Reviewer Confidence:**

3: Pretty sure, but there's a chance I missed something. Although I have a good feel for this area in general, I did not carefully check the paper's details, e.g., the math, experimental design, or novelty.

**Typos Grammar Style And Presentation Improvements:**

I would suggest that in the introduction, you include a paragraph where you clearly state your contributions.

---

> ### Author Rebuttal · Authors · 2023-08-28
>
> Thank you for the insightful comments and for your appreciation of our dataset as a resource “valuable for future work on empathic analysis” as well as your interest in our user study’s efforts to “conduct personalized evaluation of the model’s ability to retrieve similar stories the user can empathize with.”
>
> 1.  _Summarization methodology_:
> * This is a great point, as summarization definitely distills the meaning of a story and could filter out valuable main event, emotion, and moral information, as you mention. We initially ran annotation on pairs of full stories, but found that each task took on average > 10 minutes and had very low agreement between annotators. Longer, more demanding tasks (e.g., reading two full stories and comparing them) can be challenging to run on MTurk due to worker attentiveness issues. Summarization drastically reduced the complexity of the task and made it more doable for annotators (average of 8.5 minutes per HIT).
> * We do not believe that summarizing filtered critical final features in our results: We ran experiments showing that there is great presence of the human rated main event, emotion, and moral in automated summaries. We found that cosine similarity of the _human annotated_ event, emotion, and moral to the ChatGPT summarized stories (using SBERT embeddings) are **0.66 for event, 0.64 for emotion, and 0.49 for moral**.  Concatenating the human-written summaries of event, emotion and moral, and comparing to the ChatGPT summaries gives us a **semantic similarity of 0.67**,  again suggesting there is high semantic overlap in human annotated features of the story and the automatically generated summaries
> * Nevertheless, we will run and include a more in-depth analysis and human evaluation of the quality of the ChatGPT summaries for the final version of the work. Ultimately, the empathic similarity labels after summarization still provided sufficient learning signal, which resulted in improved performance not only across automatic metrics, but also in people’s actual empathic responses during the user study.
>
> 2. _Interrater Agreement_:
> * We aggregate annotations by averaging between the 2 raters. Averaging provides a good learning signal for our ultimate task of predicting empathic similarity, as most common disagreements are at most 1 likert point away (73% of points are at most 1 distance away). We will add agreement measures with this softer constraint in the paper to make this clearer.
>
> * Inter-annotator agreement is on average 0.82 for PPA and 0.21 for similarity ratings of empathic similarity, event, emotion, and moral similarity. We mention in lines 277-286 that this is in-line with other subjective, affect based tasks (Sap et al. 2017; Rashkin et al. 2018). For example, tasks on annotating power and agency (Sap et al. 2017) achieved .56 and .51 PPA and 0.34 KA and other work on annotating basic emotions and Maslow's needs achieved 0.7 - 0.91 PPA and 0.29 - 0.32 KA (Rashkin et al. 2018). Given the difficulty of our task (reading longer stories and projecting the mental state of 2 characters), our agreement is in line with prior work.
>
> 3. _Human evaluation of empathy reasoning task_:
> * Thank you for your point on providing additional human evaluation of the empathy reasoning task. We could indeed conduct a human evaluation on the summaries, but the primary aim of our human study was to show how empathic similarity makes people feel more connected, indicating the larger downstream impact of the main task, so we resorted to automatic metrics for the auxiliary task. For the final work, we will consider running human evaluations to supplement the automatic evaluations in Table 5.
>
> 4.  _Clarifying contributions_:
> * Thank you so much for your detailed feedback! We will make sure to include a list summarizing contributions at the end of the introduction. To summarize, the core contributions of this work are:
>     * A new framework/design approach for defining empathic similarity backed by social psychology and narrative psychology theory on empathic similarity that is validated by the results of a user study.
>     * The collection and annotation of a new dataset of personal stories annotated with story events, emotions, and morals, as well as empathic similarity between pairs of stories.
>     * A novel task and first evaluation benchmark for quantifying empathic similarity between personal stories, comparing current approaches, including fine-tuning and prompting LLMs.
>     * Automatic evaluation showing significant improvements of finetuning on our dataset for the new task of empathic similarity prediction.
> * A user study assessing the effect our model has on retrieving stories users empathize with __significantly more__ than off the shelf retrieval, improving empathy towards people’s stories in the real world.
> We will make sure to include this in the camera ready to make the contributions clearer.

---

### Official Review · Reviewer_RNMN · 2023-08-11

**Soundness:** 4

**Excitement:**

4: Strong: This paper deepens the understanding of some phenomenon or lowers the barriers to an existing research direction.

**Paper Topic And Main Contributions:**

The paper introduces a new task of similarity detection in personal stories based on the extend of empathy people have to each other stories instead of using raw semantic similarity.

**Reasons To Accept:**

The paper introduces a new and interesting task in conjunction of NLP and Social Psychology to identify the similarity of two texts, specifically two personal stories, based on the extend of their narrator empathy to the other story.

The paper introduces a new dataset for this task., which is carefully analyzed and they also provide the first benchmark on this dataset.

The author(s) formulated the task with mathematical notations which helps in understanding the task and further development of the task by new research teams.

They did human evaluations to asses their model performance.

**Reasons To Reject:**

My concerns have been addressed in the rebuttal.

**Reproducibility:**

4: Could mostly reproduce the results, but there may be some variation because of sample variance or minor variations in their interpretation of the protocol or method.

**Reviewer Confidence:**

3: Pretty sure, but there's a chance I missed something. Although I have a good feel for this area in general, I did not carefully check the paper's details, e.g., the math, experimental design, or novelty.

**Typos Grammar Style And Presentation Improvements:**

In section 7, further relation work, in paragraph 2, line 3, you used two usings.

---

> ### Author Rebuttal · Authors · 2023-08-28
>
> We are thankful for the reviewer’s appreciation of our contributions in “a new and interesting task in conjunction of NLP and Social Psychology,” as well as our “new dataset for this task, which is carefully analyzed” and “first benchmark on this dataset.”
>
> 1.  _Effectiveness of work relative to other related works_:
> * We apologize that the structure of the paper and the related work muddled the contributions of our work relative to prior literature.
> Prior works either focus on predicting empathy [1,2], improving empathy towards AI agents through improving empathetic dialogue generation [3], or understanding patterns of empathy online (Roshanei et al. 2019; Sharma et al. 2020) [4]. These existing tasks are typically one-way, looking at expression of empathy in utterances or generating more empathetic responses given a context. Our task is novel relative to these works because it focuses on pairwise empathy and modeling interpersonal dynamics to improve human-human connection. Furthermore, we validate our dataset, framework, and task in a user study with positive results and also briefly look at patterns of empathy (Roshanei et al. 2019; Sharma et al. 2020) [4] in section 6.3 of the paper. We will be sure to make this clearer in the camera ready.
> * Prior works (Sharma et al. 2020) [4] developed frameworks for empathy or social support that are psychologically inspired, but did not conduct real-world studies to scientifically validate frameworks or measure changes in cognitive and affective states of people.  As discussed in lines 523-526 of our paper, works that conducted studies to assess effect of story matching (Dinakar et al. 2012) report that only 58.3% found the matched story to be helpful if provided to the narrator of the original story. In contrast, we found that our model retrieved stories, users empathized with _significantly_ more than standard retrieval.
> * Since this is a new task, as a first evaluation benchmark for this task, making direct comparison of metrics is challenging — however we do show significance of improvements in both automatic evaluation and our user study. Relative to prior works, our methodology in data collection, annotation, and model training is similar and uses current, state-of-the-art trends in quantitative NLP (including comparisons of fine-tuning LLMs and few-shot prompting of very large language models). In addition, as part of our development, we also evaluated other models like GPT-2 and Flan-T5 and conducted extensive ablations and selected and presented only the best performing models (BART, GPT-3/3.5). We will make sure to include this in the camera ready.
>     * __Dataset__: While prior works (Sharma et al. 2020; Saldias et al. 2020) [5] have personal story datasets, none of them have empathy annotations suitable for our task. As such, we collect additional personal stories and combine these datasets into our evaluation.
>
>     * __Model training and automatic evaluation__: For fine-tuning, we used similar approaches to Sharma et al. 2020, and we used the same few shot prompting setup as Sap et al. 2022. Our automatic evaluation metrics are standard for semantic similarity and text generation tasks.
> * We will restructure the related work earlier in the paper, as you mentioned, and all of the above points about how we position our contributions relative to related work.
>
> 2.  _Typos and presentation improvements_: Thank you for your detailed feedback! We will correct the typos you pointed out.
>
> [1] Lahnala, A., Welch, C., and Flek, L. 2022. CAISA at WASSA 2022: Adapter-Tuning for Empathy Prediction. In Proceedings of the 12th Workshop on Computational Approaches to Subjectivity, Sentiment & Social Media Analysis, pages 280–285, Dublin, Ireland. Association for Computational Linguistics.
>
> [2] Buechel, S., Buffone, A., Slaff, B., Ungar, L., and Sedoc, J. 2018. Modeling Empathy and Distress in Reaction to News Stories. In Proceedings of the 2018 Conference on Empirical Methods in Natural Language Processing, pages 4758–4765, Brussels, Belgium. Association for Computational Linguistics.
>
> [3] Rashkin, H., Smith, E. M., Li, M., Boureau, Y. 2019. Towards Empathetic Open-domain Conversation Models: A New Benchmark and Dataset. In Proceedings of the 57th Annual Meeting of the Association for Computational Linguistics, pages 5370–5381, Florence, Italy. Association for Computational Linguistics.
>
> [4] Bao, J., Wu, J., Zhang, Y., Chandrasekharan, E., and Jurgens, D. 2021. Conversations Gone Alright: Quantifying and Predicting Prosocial Outcomes in Online Conversations. In Proceedings of the Web Conference 2021 (WWW '21). Association for Computing Machinery, New York, NY, USA, 1134–1145.
>
> [5] Giorgi, S., Zhao, K., Feng, A. H., & Martin, L. J. (2023). Author as Character and Narrator: Deconstructing Personal Narratives from the r/AmITheAsshole Reddit Community. Proceedings of the International AAAI Conference on Web and Social Media, 17(1), 233-244.

---

### Official Review · Reviewer_rDq9 · 2023-08-17

**Soundness:** 4

**Excitement:**

4: Strong: This paper deepens the understanding of some phenomenon or lowers the barriers to an existing research direction.

**Missing References:**

One glaring omission in the reference list is work done by David Jurgens et al. on sympathy and empathy in online fora.

Zhou, N. and Jurgens, D., 2020, November. Condolence and empathy in online communities. In Proceedings of the 2020 Conference on Empirical Methods in Natural Language Processing (EMNLP) (pp. 609-626).

Lahnala, A., Welch, C., Jurgens, D. and Flek, L., 2022. A critical reflection and forward perspective on empathy and natural language processing. arXiv preprint arXiv:2210.16604.

Bao, J., Wu, J., Zhang, Y., Chandrasekharan, E. and Jurgens, D., 2021, April. Conversations gone alright: Quantifying and predicting prosocial outcomes in online conversations. In Proceedings of the Web Conference 2021 (pp. 1134-1145).

Other more NLP-focused work could also be added to strengthen the arguments made by the authors:

Lahnala, A., Welch, C. and Flek, L., 2022, May. CAISA at WASSA 2022: Adapter-tuning for empathy prediction. In Proceedings of the 12th Workshop on Computational Approaches to Subjectivity, Sentiment & Social Media Analysis (pp. 280-285).

Kumano, S., Ishii, R. and Otsuka, K., 2017, October. Comparing empathy perceived by interlocutors in multiparty conversation and external observers. In 2017 Seventh International Conference on Affective Computing and Intelligent Interaction (ACII) (pp. 50-57). IEEE.

Boukricha, H., Wachsmuth, I., Carminati, M.N. and Knoeferle, P., 2013, September. A computational model of empathy: Empirical evaluation. In 2013 Humaine Association Conference on Affective Computing and Intelligent Interaction (pp. 1-6). IEEE.

**Paper Topic And Main Contributions:**

This paper presents an empathic similarity dataset with pairwise annotations for 2000 pairs of stories from a pool of 1500 stories. The authors show that their finetuned model is better aligned with human interpretations of empathic similarity than previous models.

**Questions For The Authors:**

Will you make the code available online?
Will the dataset be available online?
Although you list many different similarity scores and model performance metrics (tables 3, 4, 5) I'm unconvinced these are very significant. Of course, fine-tuning will make the model more domain-specific and this is where I think you should bring the results from section 6 to show that these are not just generic domain-adaptation improvements but actually useful from a real-world perspective where your model is the best at detecting empathic similarity (which I do believe it probably is). Could you present this disconnect better somehow to improve the narrative of your paper?

**Reasons To Accept:**

The methodology is overall sound and some of the concepts introduced in this paper are fairly novel as well useful and interesting. The authors also manage to discuss the matter from an interdisciplinary perspective bringing in psychology and social studies work.

The dataset is also likely to be useful to other researchers specifically in empathy and intimacy-related projects, but also sentiment analysis and summarization projects.

Additionally, to the delight of CS people everywhere, the authors evaluate their model using BLEU, ROUGE, and METEOR.

**Reasons To Reject:**

This paper is a little confused about what it wants to be and is in some ways two papers merged into one as exemplified by section 6 which is a standalone study with its own hypothesis, previous work, data, and method as well as results section and is followed by section 7 "Further related work" just before the "Conclusions" section. Although these issues are structural they confound the core message of the paper which should be the pairwise empathic dataset.

I am also not entirely convinced of the results and their significance. At the very least, the improvements of the model presented here should be discussed more in-depth in the text. As it currently stands the link between the results in table 3-5 and figures 3 and 4 are confusing at best.

**Reproducibility:**

4: Could mostly reproduce the results, but there may be some variation because of sample variance or minor variations in their interpretation of the protocol or method.

**Reviewer Confidence:**

3: Pretty sure, but there's a chance I missed something. Although I have a good feel for this area in general, I did not carefully check the paper's details, e.g., the math, experimental design, or novelty.

**Typos Grammar Style And Presentation Improvements:**

I'm not entirely certain that the flow of the paper is the best. I understand the choice to place section 6 where it is, but it could very well be merged into previous sections as another hypothesis. Currently, section 6 reads like its own short paper so perhaps the authors could consider redistributing the information of section 6 into the other sections (intro, previous work, data & method, results etc).

Similarly, it is strange to see section 7 "Further related work" as the next to last chapter just before "conclusions". I suggest that baking section 6 into the whole paper includes moving section 7 as a subsection to section 2. Otherwise, the paper is just too disjointed and reads as if trying to cram two papers into one. If your paper is not accepted, you could consider resubmitting it elsewhere as two short papers rather than one long paper.

---

> ### Author Rebuttal · Authors · 2023-08-28
>
> We are grateful for the reviewer’s positive comments in valuing our contributions in interdisciplinary approaches as well as the usefulness of the dataset for future work in empathy. Thank you for your comments on how “some of the concepts introduced in this paper are fairly novel as well as useful and interesting.” We also appreciate the additional references provided — we will include these in the camera ready!
>
> 1.  _Clarifying core contributions and flow of the paper and role of user study_:
> * We apologize that the structure and flow of the paper muddled the contributions of the paper, and specifically the role of our human study. In summary, the core contributions of the work are (1) a new framework defining empathic similarity based on social and narrative psychology (2) a new dataset of personal stories annotated with story events, emotions, morals, and empathic similarity between pairs of stories, (3) automatic evaluation on our new task of empathic similarity retrieval and (4) a real-world user study demonstrating that our model significantly improves empathy towards people’s stories compared to off-the-shelf retrieval systems (see rebuttal 4 for full detailed list of contributions).
>
>     The user study in this work is crucial to showcase the promising real-world impact of our new task. Prior work's versions of human evaluations (Sharma et al. 2020) [1, 2] are humans verifying or ranking model outputs based on inputs from dev/test data. This provides a valuable signal of model quality, but isn't representative of how a model could be used in real-world applications (due to input distribution mismatch and lack of personal investment in the task). Our human evaluation is a full user study to see how the model performs given a story that the users told themselves, which is much more aligned with real-world applications. We demonstrate the applicability of the task to improve empathy towards retrieval of human stories, and because of this extensive human evaluation, we included separate hypotheses. The user study fits the overall narrative of why we collected this dataset, how it was used to develop this task, and why this task matters in the real world. We will be sure to improve the structure and narrative of the paper to make this clearer.
> * As you also mentioned, showing actual real-world usefulness of the models is important for the narrative of the paper, which is why we show in our human evaluation with 150 participants that stories retrieved using our model significantly improve empathy towards selected stories over current baselines such as SBERT story retrieval. This is particularly important given the highly personalized aspects of the study — participants wrote a story from their own life, and our approach, motivated by social psychological theory, was able to capture more empathically relevant features of their personalized stories and result in meaningful impact on their empathy levels. We will make sure to tie this discussion into the automatic evaluation results in the camera ready.
>
> 2. _Significance of results_:
> * Good point on domain-specificity of the fine tuning task! The improvements we reported are statistically significant (***p* = 0.02** for BART finetuning improvements based on McNemar’s test on the accuracy scores, similarly we obtain ***p* = 0.0006** comparing Spearman’s correlation with/without fine-tuning on BART using Fisher’s z-transformation). We will make this clearer in the camera ready by indicating statistically significant elements in Table 4.
>
> 3.  _Related work_:
> * Thank you for your detailed feedback, we will correct all of the structural elements of the paper and tighten up the related work earlier in the paper. In the camera ready, we will rearrange the related work and motivate our work in relation to prior work better, as suggested. In summary, few prior works on text-based empathy have looked at modeling empathy in two-way interpersonal settings for human-to-human connection (most focus on detecting empathy or generating empathetic utterances), and even fewer of these works have shown tangible outcomes in human studies. The authors believe this task is crucial given the increased loneliness and apathy between people over time [3]. With increasing polarization, personal experiences are a fundamental way people connect, yet existing social recommendation isn’t targeted for human-human connectivity and empathy. Empathically encoded story embeddings could be useful for a variety of NLP tasks, including retrieval, text generation, dialogue, and translation, for example in the following settings:
>     *  Using empathic reasoning to incorporate story retrieval in dialogue generation.
>     * Generating stories that users resonate with more in conversational AI
>     * Extending this work to multilingual settings and better understand translating experiences in ways that preserve empathic meaning
>     * Better understand cognitive insights, such as linguistic patterns of emotional communication
>     * Applications/building interactions that foster story sharing across geographic, ethnic, and cultural bridges, such as developing better social media recommendation or personalization.
>
> * We will make sure to ground these motivations in the related work and help support the significance of our results.
>
> 4. _Reproducibility of work_:
> * The dataset we created is unique and adds a new contribution to the field (as mentioned in footnote 1 on page 2). The dataset will be made publicly available with all annotations, and we will release all code including annotation templates, pre-processing of data, model fine tuning, and evaluation code (anything used to replicate the entire study). We will also make all user study interfaces publicly available to allow replication of the user study.
>
> [1] Zhou, N. and Jurgens, D. 2020. Condolence and Empathy in Online Communities. In Proceedings of the 2020 Conference on Empirical Methods in Natural Language Processing (EMNLP), pages 609–626, Online. Association for Computational Linguistics.
>
> [2] Bao, J., Wu, J., Zhang, Y., Chandrasekharan, E., and Jurgens, D. 2021. Conversations Gone Alright: Quantifying and Predicting Prosocial Outcomes in Online Conversations. In Proceedings of the Web Conference 2021 (WWW '21). Association for Computing Machinery, New York, NY, USA, 1134–1145.
>
> [3] Buecker S., Mund M., Chwastek S., Sostmann M., and Luhmann M. Is loneliness in emerging adults increasing over time? A preregistered cross-temporal meta-analysis and systematic review. Psychol Bull. 2021 Aug;147(8):787-805.

---

### Official Review · Reviewer_g6sS · 2023-08-17

**Soundness:** 4

**Excitement:**

4: Strong: This paper deepens the understanding of some phenomenon or lowers the barriers to an existing research direction.

**Paper Topic And Main Contributions:**

The paper is a good attempt to enable social-emotional correlation between people based on their stories.  The dataset of 2000 stories is constructed. 150 participants feedback on the story is also created and compared with the model. The stories are classified into main, empathy and moral events.

**Questions For The Authors:**

The impact of a personal story and its similarity is useful to get some valid conclusion ?? though it is not in the scope, the impact needs to be calculated in order to find the success of the method.  Please write a paragraph about this point in a paper.

**Reasons To Accept:**

The paper is a good attempt to enable social-emotional correlation between people based on their stories.  The dataset of 2000 stories is constructed. 150 participants feedback on the story is also created and compared with the model. The stories are classified into main, empathy and moral events.

**Reasons To Reject:**

The concerns are addressed in the revision.

**Reproducibility:**

3: Could reproduce the results with some difficulty. The settings of parameters are underspecified or subjectively determined; the training/evaluation data are not widely available.

**Reviewer Confidence:**

4: Quite sure. I tried to check the important points carefully. It's unlikely, though conceivable, that I missed something that should affect my ratings.

---

> ### Author Rebuttal · Authors · 2023-08-28
>
> We appreciate your  insightful comments, and for acknowledging the efforts of our work in modeling social-emotional relevance of stories with respect to empathy.
>
> 1. _Clarification of impact of our method on empathic similarity prediction and empathy summaries_:
> * We see that the impact of our method in retrieving emphatically similar stories might not be clear.  For metric-wise impact, we show through automatic evaluation that *models fine tuned on our dataset significantly outperform baselines* in empathetically similar story retrieval. We also find that for BART models, fine tuning improvements, as measured with McNemar’s test on the accuracy scores and Fisher’s transformation on correlations, are significantly higher than baselines (***p*= 0.02, *p*= 0.0006** respectively; results seen in Table 4).  In lines 374-386, we further elaborate on the impact fine-tuning with our dataset has across a variety of automated metrics for semantic similarity and retrieval. We will clarify this in the final version.
> * A core contribution of our work is showing the _real-world impact of our model on user’s empathy_ towards stories selected by our model. In our human evaluation, we report Cohen’s effect size (*d*=0.26) and statistical significance (*p*<0.01) showing that *our methods retrieve stories that real world users empathize significantly better with  (Section 6.3).*
>
> * We thank the reviewer for mentioning evaluating the impact of the empathy summaries. We provide automated evaluation metrics for the quality of these summaries using BLEU, ROUGE, METEOR, and BertScore in Table 5. For the empathy reasons summarization task, the input is a single personal story, and the outputs are summaries of the (1) main event of the story, (2) emotional trajectory of the story, and (3) the overall takeaway/moral of the story. These stories also may contain conversational elements, which are also included in the output summaries depending on their relevance.
>
> 2.  _Exploration of more models and datasets_:
>
> * Currently, no existing datasets containing empathy relationships between personal stories exist. As such, we searched for the most relevant literature (Sharma et al. 2020; Saldias et al. 2020) [1] and used stories from these datasets, as well as our own collected data, for our final dataset.
>
> * We used state-of-the-art models (gpt-3.5, gpt-3, BART, and SBERT). In preliminary experiments, we also tested other transformer-based models (ie. GPT-2, Flan-T5), but found their performance to be worse than the BART model, so we did not include these in paper. We will make a note in the paper.
>
> 3. _Personal story conversational elements_:
> * Personal stories contain conversational elements (e.g., quotes), and are typically included in the summary when the conversation is important enough to affect the overall trajectory of the story.  We will make a note of this in the camera ready.
>
> 4. _Participant selection_:
> * We selected participants over the age of 18 who are from predominantly English-speaking countries (US and UK), as approved by our institution’s IRB. This will be clarified in the  camera ready.
>
> 5. _Clarifying methodology_:
> * “Why is cosine similarity used?” As shown in (Reimers et al. 2017), cosine similarity is an effective distance function for semantic similarity tasks, which is most analogous to our new task of empathic similarity prediction. It is also used generally for text generation metrics (Zhang et al. 2020). There is more literature on classical word embeddings that supports this [2].  As such, we incorporated this metric as our primary means of comparing empathic similarity.
> * “Line No 301,302 says the score is less for dissimilar. How?” - For empathically similar stories, we aim to have the model return a high similarity score, pulling empathically similar stories (as rated by human labelers) closer in the embedding space. Conversely, for dissimilar stories, one would have a smaller score. We will clarify this in the camera ready.
> * “explain fig 2 working with an example.”
> An example for each element would be:
>     - Story A example: “Once upon a time, there was a curious dog named Biscuit who loved exploring. While playing fetch, he accidentally stumbled on a hidden treasure chest filled with delicious bones. From that moment on, Biscuit was always eager to embark on new adventures.”
>     - Annotator A summarizes event (“Biscuit stumbled on a treasure chest”), emotion (“Biscuit is excited and curious when he finds the chest”), and moral (“curiosity can lead to unexpected positive discoveries”).
>     - Story B example: “There was an imaginative child named Lily who loved to explore. She was curious about an abandoned treehouse in her neighborhood and went to explore. There, she found a kitten and they became good friends.”
>     - Annotator B summarizes event (“While exploring a treehouse, Lily finds a kitten.”), emotion (“Lily feels curious about the treehouse and happy when she makes a new friend.”), and moral (“crossing uncharted territories can lead to new joys”)
>     - Concatenating encode(event_A + emotion_A + moral_A) compared with encode(event_B + emotion_B + moral_B). Based on the sampling strategy described in Lines 234-248, we sample the pair (Story A, Story B) to provide to raters.
>     - Annotator C decides how empathically similar  story A and story B are to one another (on scale of 1-4).
>
> [1] Giorgi, S., Zhao, K., Feng, A. H., & Martin, L. J. (2023). Author as Character and Narrator: Deconstructing Personal Narratives from the r/AmITheAsshole Reddit Community. Proceedings of the International AAAI Conference on Web and Social Media, 17(1), 233-244.
>
> [2] Lahitani, A. R., Permanasari, A. E., and Setiawan N. A. Cosine similarity to determine similarity measure: Study case in online essay assessment. 2016 4th International Conference on Cyber and IT Service Management, Bandung, Indonesia, 2016, pp. 1-6.

---

### Meta-Review · Area_Chair_4EQx · 2023-09-18

**Recommendation:** 5

**Metareview:**

This paper proposes a framework operationalizing empathic similarity (i.e., similarity between personal stories based on empathic resonance). It contributes 1) a dataset of personal stories annotated with features from the framework and pairs of stories annotated with empathic similarity scores, 2) a model fine-tuned to compute empathic similarity of story pairs, and 3) a user study showing that participants empathize more with model-retrieved stories.

There is strong consensus from the reviewers that this paper presents a valuable contribution. They appreciate that the paper draws on work outside NLP (e.g., social psychology) to engage meaningfully with empathy and its expression in narratives, and find the concepts and task introduced to be new and interesting. They find the study to be well-scoped and the methodology sound, and appreciate the concepts’ and dataset’s likely utility not only for researchers working directly on empathy-related questions, but also for researchers working on sentiment, summarization, and other tasks that would benefit from engagement with the expression of personal experiences.

---

### Decision · Program_Chairs · 2023-10-07

**Decision:**

Accept-Main

**Comment:**

This paper proposes a framework operationalizing empathic similarity (i.e., similarity between personal stories based on empathic resonance). It contributes 1) a dataset of personal stories annotated with features from the framework and pairs of stories annotated with empathic similarity scores, 2) a model fine-tuned to compute empathic similarity of story pairs, and 3) a user study showing that participants empathize more with model-retrieved stories.

There is strong consensus from the reviewers that this paper presents a valuable contribution. They appreciate that the paper draws on work outside NLP (e.g., social psychology) to engage meaningfully with empathy and its expression in narratives, and find the concepts and task introduced to be new and interesting. They find the study to be well-scoped and the methodology sound, and appreciate the concepts’ and dataset’s likely utility not only for researchers working directly on empathy-related questions, but also for researchers working on sentiment, summarization, and other tasks that would benefit from engagement with the expression of personal experiences.